# A Scalable Approach to Internet of Things and Industrial Internet of Things Security: Evaluating Adaptive Self-Adjusting Memory K-Nearest Neighbor for Zero-Day Attack Detection

**DOI:** 10.3390/s25010216

**Published:** 2025-01-02

**Authors:** Promise Ricardo Agbedanu, Shanchieh Jay Yang, Richard Musabe, Ignace Gatare, James Rwigema

**Affiliations:** 1African Centre of Excellence for Internet of Things, University of Rwanda, Kigali P.O. Box 4285, Rwanda; j.rwigema@ur.ac.rw; 2Global Cybersecurity Institute, Rochester Institute of Technology, Rochester, NY 14623, USA; 3Institute for Informatics and Applied Technology, Gonzaga University, Spokane, WA 99258, USA; yangj@gonzaga.edu; 4College of Science and Technology, University of Rwanda, Kigali P.O. Box 4285, Rwanda; r.musabe@ur.ac.rw (R.M.); i.gatare@ur.ac.rw (I.G.)

**Keywords:** iInternet of Things, Industrial Internet of Things, cybersecurity, online machine learning, zero-day attacks, intrusion detection system

## Abstract

The Internet of Things (IoT) and Industrial Internet of Things (IIoT) have drastically transformed industries by enhancing efficiency and flexibility but have also introduced substantial cybersecurity risks. The rise of zero-day attacks, which exploit unknown vulnerabilities, poses significant threats to these interconnected systems. Traditional signature-based intrusion detection systems (IDSs) are insufficient for detecting such attacks due to their reliance on pre-defined attack signatures. This study investigates the effectiveness of Adaptive SAMKNN, an adaptive k-nearest neighbor with self-adjusting memory (SAM), in detecting and responding to various attack types in Internet of Things (IoT) environments. Through extensive testing, our proposed method demonstrates superior memory efficiency, with a memory footprint as low as 0.05 MB, while maintaining high accuracy and F1 scores across all datasets. The proposed method also recorded a detection rate of 1.00 across all simulated zero-day attacks. In scalability tests, the proposed technique sustains its performance even as data volume scales up to 500,000 samples, maintaining low CPU and memory consumption. However, while it excels under gradual, recurring, and incremental drift, its sensitivity to sudden drift highlights an area for further improvement. This study confirms the feasibility of Adaptive SAMKNN as a real-time, scalable, and memory-efficient solution for IoT and IIoT security, providing reliable anomaly detection without overwhelming computational resources. Our proposed method has the potential to significantly increase the security of IoT and IIoT environments by enabling the real-time, scalable, and efficient detection of sophisticated cyber threats, thereby safeguarding critical interconnected systems against emerging vulnerabilities.

## 1. Introduction

The Internet of Things (IoT) and the Industrial Internet of Things (IIoT) have revolutionized how industrial systems are designed, operated, and managed, increasing efficiency, reliability, and flexibility. However, these systems’ increased connectivity and interconnectivity have also presented new and complex security challenges, as they are more vulnerable to cyber-attacks that can disrupt critical infrastructure and compromise sensitive data [1].

In recent years, there has been a surge in the number of cyber-attacks targeting industrial systems, raising concerns about the security of the IIoT [2,3]. Cyber-attacks pose a significant threat to the IIoT and IoT, leading to several studies trying to solve this problem [4,5,6,7]. Cyber-attacks like malware can infiltrate an IIoT system and remain undetected for long periods, leading to significant economic losses, safety risks, and potential environmental damage. Malware can also spread rapidly across interconnected systems, making detecting and containing the attack even more challenging. Designing effective and efficient systems for detecting these attacks in the IIoT is essential for mitigating these risks.

The proliferation of IoT devices across diverse domains has significantly increased the frequency and complexity of cyber-attacks targeting these networks [8]. A particularly concerning threat is the emergence of zero-day attacks, which exploit previously unknown vulnerabilities before they can be detected and mitigated [8]. Traditional signature-based intrusion detection systems (IDSs) struggle to detect such novel attacks effectively, as they rely on prior knowledge of attack patterns [9]. Researchers have explored machine learning (ML) techniques to develop more adaptive and intelligent IDSs for IoT environments [10].

Most intrusion detection methods are not well suited to the unique requirements of the IoT and IIoT because they are computationally expensive, need to be retrained in an off-production mode to adapt to new attack trends, and cannot detect zero-day attacks. Additionally, most intrusion detection methods are based on offline ML techniques. Offline ML-based approaches are often too complex and computationally expensive to be implemented in computationally constrained devices like the IoT and IIoT. Secondly, ML-based intrusion detection systems are not real-time adaptive. Because they are trained using historical data, they cannot adapt to the dynamic nature of the IIoT ecosystem. Finally, updating offline models requires the models to be retrained on new data, which introduces delays between when a new attack sample appears and when the model is updated to detect those new samples. Due to the critical nature of IIoT systems, this short delay can make these systems vulnerable to zero-day attacks, which may end up causing damage to these production systems.

Based on the above-mentioned drawbacks of current systems that detect cyber-attacks in the IoT and IIoT, it is important to use a more effective and efficient solution to detect attacks within these ecosystems. This solution should be able to detect zero-day attacks and be computationally inexpensive. Online ML models, in particular, offer a promising approach for detecting zero-day attacks in resource-constrained IoT systems [11]. By continuously learning from network traffic data, these models can adapt to evolving attack patterns without the need for extensive retraining [11].

This paper proposes an IDS for the IoT and IIoT using an online ML technique to build an adaptive IDS capable of identifying previously unseen threats in real time, thereby enhancing the security of IoT and IIoT networks.

This article is a revised and expanded version of a paper entitled “An Online Adaptive Approach to Detecting Zero-day Attacks in IoT and IIoT Systems” [12], which will be presented at a workshop at the IEEE Global Communications Conference in Cape Town, South Africa from 8 to 12 December 2024. This expanded version contains about 60% new work, which includes the addition of a new dataset used for validating this study. In the workshop paper, we used a generative adversarial network (GAN) to introduce some synthetic attacks into the respective datasets. However, this work presents two types of zero-attacks, the first being unseen attack classes and the second being synthetic attacks created using a conditional tabular generative adversarial network (CTGAN). The expanded work also includes an expanded validation that focuses on scalability, false positive and detection rate, performance under drift, resource utilization, ablation study, and expanded statistical analysis. This study also analyzed the complexity of the proposed classifier.

The contributions of this research can be summarized as follows:Development of an Adaptive Online k-Nearest Neighbors (kNN) with Self Adjusting Memory (SAM) Classifier: Introduces the proposed classifier, an enhanced version of the SAMKNN classifier tailored for online learning scenarios. Our proposed adaptive SAMKNN dynamically adjusts its memory allocation between short-term memory (STM) and long-term memory (LTM) based on real-time performance metrics, enabling it to effectively handle evolving data distributions inherent in IoT and IIoT environments.Dynamic Memory Allocation Based on Performance Metrics: Implements a mechanism where the proportion of memory allocated to STM and LTM is dynamically adjusted based on the classifier’s recent performance. By monitoring metrics such as accuracy over a sliding window, the system can allocate more resources to STM during periods of rapid concept drift and shift toward LTM when data distributions stabilize.Efficient Memory Management for Resource-Constrained Environments: Employs Python’s double-ended queue (deque) for managing STM, allowing for efficient append and pop operations with fixed maximum lengths. Additionally, it uses NumPy arrays for LTM to facilitate rapid computations and memory compression, making the system suitable for the resource-constrained nature of IoT and IIoT devices.Comprehensive Performance Evaluation and Memory Adjustment Strategies: Utilizes both performance-based increases and decreases in memory allocation, supported by predefined thresholds and a cool-down mechanism. This bidirectional adjustment ensures that the classifier can balance adaptability with stability, maintaining high accuracy even as threat patterns evolve.

The remainder of this paper is organized as follows: Section 2 gives an overview of the related works considered in this study. In Section 3, we present our proposed methodology. Section 4 presents the experimental design of this study, with the experimental results presented in Section 5 and discussions presented in Section 6. This study is concluded in Section 7.

## 2. Related Work

Detecting zero-day attacks in IoT environments has been an area of concern for many years. As such, several works have been carried out in the quest to solve this problem. Popoola et al. [13] proposed an optimal deep neural network (DNN) architecture to detect zero-day attacks in IoT systems. The DNN was used with federated ML, where the federated algorithm is used to aggregate local model updates. Similarly, ref. [14] developed a framework based on federated learning to detect zero-day botnet attacks in IoT systems. As part of the contributions of their work, ref. [14] also developed a novel aggregation algorithm that handles model aggregation better in IoT systems. Due to the limited computational power, memory, and battery life of IoT devices, training and updating complex DNN models or performing federated learning on these devices can be resource-intensive.

Hairab et al. [15] evaluated the performance of convolutional neural networks (CNNs) in detecting zero-day attacks in the IoT using two regularization techniques. The authors reported that the use of regularization techniques increased the performance of the proposed system, with an ability to detect zero-day attacks. Similarly, ref. [16] also proposed an anomaly detection system for IoT systems using a CNN with regularization methods. In another study, ref. [17] proposed an IDS based on a CNN that could detect seen and unseen attacks in the Internet of Vehicles ecosystem. The proposed IDS was developed at the data processing layer of the Internet of Vehicles ecosystem. The rationale behind the development of the IDS at this layer is to speed up the rate of detection of the IDS. The authors also compared the proposed method to an SVM and RF, of which the proposed method recorded a higher detection accuracy. IoT devices are highly heterogeneous, varying in terms of hardware, software, and functionality. A CNN model trained on one type of device or environment may not perform well on others.

Federated learning has emerged as a promising approach to enhance detection capabilities against zero-day attacks by sharing attack information among multiple IoT networks [14,18]. By utilizing federated learning, researchers have developed frameworks that improve the detection of zero-day attacks while preserving user privacy and minimizing communication overhead [19]. These frameworks leverage deep learning (DL) algorithms to detect and resist botnet attacks in IoT networks, showcasing superior performance in identifying new and evolving threats compared to traditional centralized approaches [14]. The use of federated learning in intrusion detection systems not only enhances detection performance but also addresses resource constraints in IoT devices, offering scalable and cost-effective solutions for early attack detection and patch creation [20]. To detect zero-day attacks, ref. [21] developed an IDS model based on federated incremental learning that aggregates knowledge from different detectors and then updates the model in an incremental manner. IoT devices often generate heterogeneous data due to differences in device types, manufacturers, and deployment environments. Aggregating models effectively in federated learning to account for this heterogeneity is complex and can lead to sub-optimal performance.

To combat zero-day attacks, a hybrid approach combining ML and DL techniques has been proposed for effective detection and mitigation [22,23,24]. By leveraging ML classifiers and deep neural networks (DNNs) trained on diverse data streams, along with deep reinforcement learning (DRL) for dynamic model selection, a robust defense mechanism against zero-day malware in IoT devices is established. This hybrid framework not only enhances detection rates but also minimizes false positives and false negatives, achieving a remarkable 99% detection rate with minimal errors [22]. Saurabh et al. [25] proposed a hybrid IDS using supervised and unsupervised ML approaches that can detect both known and unknown attacks. Hybrid models combining ML, DL, and DRL are computationally intensive. IoT devices often have limited processing power, memory, and battery life, making it challenging to run these complex models locally.

Research has shown the effectiveness of ensemble learning in detecting zero-day attacks in IoT networks [14,26,27]. By combining different base anomaly detectors using conventional ML algorithms, ensemble learning can provide highly accurate zero-day attack detection even without prior labeled attack data. Moreover, the use of ensemble learning, particularly with random forest (RF) and extreme gradient boosting (XGB) algorithms, has been identified as top-performing in detecting zero-day attacks in IoT systems, outperforming previous methods and enhancing the performance of machine learning models [27]. In an earlier study, ref. [28] developed an ensemble classifier using processed data packets that can detect anomalies and protect IoT systems against zero-day attacks. Additionally, leveraging ensemble learning in a federated framework for IoT networks can achieve superior model aggregation without compromising user privacy, showcasing its efficacy in zero-day botnet attack detection [14]. Ahmad et al. [29] used an ensemble of DL classifiers to build an IDS. The proposed IDS is trained using four benchmark datasets while testing the model with unknown attacks to validate the system’s performance in detecting zero-day attacks. Ensemble learning models, particularly those involving multiple algorithms like RF and XGB, can be computationally intensive. This complexity increases the resource requirements, which can be problematic for resource-constrained IoT devices.

Farrukh et al. [30] focused on proposing a framework based on packet-level data by extracting spatial and temporal patterns from network traffic. The authors also used stacking and sub-clustering techniques to help to effectively detect unknown attacks. Ref. [27] proposed a computational framework that includes feature selection through fuzzification. The authors reported that RF and extreme gradient boosting (XGB) were the top-performing algorithms that could detect zero-day attacks. In another study, ref. [31] used online reinforcement learning (RL) to propose a framework that learns the correct moving target defense (MTD) to mitigate heterogeneous zero-day attacks in single-board computers (SBCs). The proposed framework works by considering behavioral fingerprinting to represent SBCs’ states and RL to learn MTD techniques that mitigate each malicious state. The results of the study show that the proposed framework mitigates all attacks except rookits while consuming less than 1 MB of storage and approximately 10% of RAM. Ref. [32] designed an IDS framework using transfer learning (TL) that could detect both known and zero-day attacks. The transfer learning model used in their study was based on a CNN. Techniques such as stacking, sub-clustering, and TL with CNNs are computationally intensive. IoT devices, especially single-board computers (SBCs), typically have limited processing power, memory, and storage.

Zero-day attacks targeting IoT devices have become a significant concern due to the vulnerabilities in interconnected devices [33]. To combat these threats, researchers have proposed innovative solutions, like a fine-grained central processing unit (CPU) security engine, μThingNet, which leverages DL and power analysis to detect unknown malware variants, with a high detection rate of 97.49% [34]. Additionally, ref. [33] used honeypot systems to detect malicious activities and analyze zero-day attacks, benefiting from filtering malicious traffic to identify and thwart attacks effectively. Furthermore, a game-theoretic approach has been introduced to strategically allocate honeypots over networks, considering the deceptive nature of attackers and the impact of zero-day vulnerabilities on defense mechanisms [35]. ML techniques have also emerged as a powerful tool for detecting zero-day attacks by analyzing patterns in network traffic and user behavior, enhancing cybersecurity defenses against these elusive threats [24]. Implementing a fine-grained CPU security engine may require specialized hardware modifications, making it difficult to deploy on existing IoT devices.

Table 1 and Table 2 provide a summary that compares existing methods to our proposed method.

## 3. Proposed Methodology

The proposed IDS is designed to identify zero-day attacks in IoT and IIoT environments by leveraging the proposed classifier. The system is structured into five key modules: preprocessing, proposed classifier, attack detection, alert and response, and performance monitoring. Each module performs a distinct function to ensure the accurate, efficient, and adaptable detection of known and unknown attacks. The following sections describe each module in detail. Figure 1 represents a block diagram of our proposed system.

### 3.1. Preprocessing Module

The initial step involves loading the dataset and performing essential preprocessing to enhance its suitability for machine learning applications. This includes the following:Feature Selection:Irrelevant and non-informative columns such as source and destination IP addresses, attack types, and redundant labels are removed to streamline the feature set. This focuses the analysis on the most pertinent variables that contribute to intrusion detection.Data Shuffling: To eliminate any inherent order or bias present in the dataset, the data are randomly shuffled. This ensures that the training and evaluation processes are not influenced by the sequence of the data, promoting a more generalized and robust model performance.Dataset Iterator Preparation: For online learning scenarios, where models are updated incrementally as new data arrive, the dataset is prepared for streaming. An iterator is created to simulate real-time data flow, allowing the model to process data sequentially in an online manner.

#### 3.1.1. Feature Extraction

In developing the proposed system, a set of features was extracted from the datasets to effectively capture the characteristics of network traffic. The feature extraction process involved selecting variables that are indicative of normal and malicious network behavior while excluding those that could introduce noise or redundancy. The dataset initially contained several columns, including IPV4_SRC_ADDR, IPV4_DST_ADDR, Attack, and Label. These columns were excluded from the feature set for the following reasons:IP Addresses (IPV4_SRC_ADDR and IPV4_DST_ADDR): The IP addresses were removed to ensure that the model focuses on behavioral patterns rather than specific network endpoints, enhancing the generalization of the IDS across different network configurations.Attack Type (Attack): The ’Attack’ column, which specifies the type of attack, was excluded to prevent the model from being biased toward known attack signatures. This exclusion is crucial for maintaining the model’s ability to detect zero-day attacks that may not conform to predefined attack types.Label (Label): The ’Label’ column serves as the target variable indicating whether a network flow is benign or malicious. It was separated from the feature set to facilitate supervised learning.

The remaining columns after the exclusion are L4_SRC_PORT, L4_DST_PORT, PROTOCOL, L7_PROTO, IN_BYTES, OUT_BYTES, IN_PKTS, OUT_PKTS, TCP_FLAGS, and FLOW_DURATION_MILLISECONDS.

The selection of features was driven by the need to balance comprehensiveness and computational efficiency, particularly given the resource-constrained nature of IoT and IIoT devices. The rationale for selecting the aforementioned features includes the following:Relevance to Intrusion Detection: The chosen features are closely related to network behavior and are effective in capturing anomalies indicative of cyber-attacks. For instance, unusual patterns in flow duration or packet counts can signal potential intrusions.Avoidance of Redundancy: By excluding IP addresses and specific attack types, the feature set minimizes redundancy and prevents the model from overfitting to particular network endpoints or known attack signatures. This approach enhances the model’s capability to generalize and detect novel threats.Efficiency in Processing: The retained features are sufficient for representing the essential aspects of network traffic without imposing excessive computational overhead, which is crucial for real-time processing in IoT and IIoT environments.

#### 3.1.2. Feature Scaling

To ensure that all features contribute equally to the learning process and to improve the performance of distance-based classifiers like kNN, feature scaling was employed. We used a standard scaler approach to standardize the feature set. This method standardizes the features by removing the mean and scaling them to unit variance. Specifically, each feature *x* is transformed using the following formula:$$x_{\mathrm{scaled}}=\frac{x-\mu }{\sigma }$$
where μ is the mean and σ is the standard deviation of the feature.

The benefits of standardizing the feature sets include;

Uniform Feature Contribution: By standardizing the features, the model ensures that no single feature dominates due to its scale, which is particularly important for algorithms sensitive to feature magnitudes.Improved Convergence: Standardization can lead to faster and more stable convergence during the training process, enhancing the overall efficiency of the model.Compatibility with Distance-Based Algorithms: For classifiers like kNN, standardized features ensure that distance calculations are meaningful and not skewed by disparate feature scales.

### 3.2. Adaptive SAMKNN Module

The core detection engine of the IDS relies on the proposed classifier, which dynamically balances between recent and historical data through two types of memory: STM and LTM.

#### 3.2.1. Short-Term Memory (STM)

STM holds recent data samples, allowing the classifier to adapt quickly to emerging attack patterns. It is essential for recognizing and responding to novel attacks that may not match historical attack signatures.

#### 3.2.2. Long-Term Memory (LTM)

LTM contains aggregated historical data, providing a stable base of knowledge for the classifier. LTM enables the system to retain context from past experiences, enhancing its ability to identify well-known attack patterns.

#### 3.2.3. Dynamic Memory Adjustment

The classifier dynamically adjusts the balance between STM and LTM based on model performance. If the accuracy improves, the LTM size increases to retain more historical data. Conversely, if performance drops, the LTM size is reduced, allowing the model to focus on recent data. The model’s performance is tracked through a sliding window of recent predictions, and memory adjustments are made when sufficient data are available, regulated using a cool-down period. The cool-down technique ensures efficient memory usage and maintains accuracy in stationary and non-stationary environments, adapting to concept drift over time.

### 3.3. Attack Detection Module

The attack detection module is responsible for identifying anomalous behavior that may indicate an attack. It processes incoming data using classifiers to distinguish between benign and malicious activity. The module leverages mechanisms like memory adjustment to handle zero-day attacks effectively. This module includes the following:

#### 3.3.1. Attack Detector

The proposed classifier processes incoming data and classifies each instance as benign or malicious based on learned patterns. In the case of zero-day attacks, the classifier’s memory adjustment mechanism enhances its capacity to detect previously unseen attack types without prior knowledge.

#### 3.3.2. Zero-Day Analysis

This component emphasizes the identification of novel attack instances by leveraging the classifier’s STM. By prioritizing recent data, the classifier can respond to emerging threats that deviate from normal patterns or known attack signatures, thereby improving the detection of zero-day attacks.

### 3.4. Alert and Response Module

This module is essential for real-time threat handling by promptly notifying administrators of detected threats and automatically responding to them to mitigate damage. It ensures a proactive approach to minimize the impact of attacks.

#### 3.4.1. Alert Manager

The alert manager triggers an alert whenever malicious activity is detected. This alert can be customized based on the threat level, providing immediate feedback to system administrators and initiating automated responses if necessary.

#### 3.4.2. Response

The system can execute pre-configured response actions, such as isolating the compromised device, restricting access to specific network resources, or blocking traffic from suspicious sources. These response mechanisms are crucial for containing and mitigating the impact of detected attacks in IoT/IIoT environments.

### 3.5. Performance Monitoring Module

The performance monitoring module continuously evaluates the system’s performance to ensure sustained accuracy and adaptability.

#### Performance Metrics

The system tracks various performance metrics, such as detection accuracy, false positives, and false negatives, to assess the efficacy of the classifier in real time.

### 3.6. Adaptive SAMKNN

In this subsection, we explain how we implemented the proposed Adaptive SAMKNN, a modification of the original SAMKNN proposed by [36].

The proposed classifier is an enhanced KNN algorithm specifically designed for dynamic streaming environments where the nature of data may evolve over time. This adaptation incorporates mechanisms for managing memory resources efficiently, dynamically adjusting between short-term and long-term memory based on observed performance, and adapting to drifts in data distribution.

The algorithm begins by initializing two memory structures, STM and LTM. STM is a memory buffer that holds recent data samples, while LTM stores older, stable data that could still be useful for classification. An initial allocation proportion $\lambda_{LTM}$ is set for LTM to control the balance between the sizes of STM and LTM. Key parameters are also initialized, including performance thresholds ($\delta_{decrease}$ and $\delta_{increase}$) that determine when the memory allocation should be adjusted based on classifier performance. Additionally, a cool-down period is set to prevent frequent adjustments and allow time for performance changes to become evident. Finally, label encoding dictionaries are prepared, which may map categorical labels to numerical values.

For each incoming data sample *x*, the algorithm follows a structured process. If the label *y* of the sample is known (i.e., the sample is labeled), the classifier first predicts the label $\hat{y}$ for *x* using both STM and LTM. This prediction is then used to update the algorithm’s accuracy metric, maintained over a performance window P, tracking how well the classifier has been performing recently.

The labeled sample (x,y) is then added to STM. However, if the size of STM exceeds a predefined threshold $M_{STM}$, the oldest samples in STM may be shifted to LTM (if such shifting is enabled). This process ensures that STM remains a reservoir of recent data without overflow while allowing LTM to accumulate samples that may still hold relevant information for classification.

The adaptive mechanism of SAMKNN comes into play after every performance window P reaches a size *W* and the cool-down period has elapsed. At this point, the classifier evaluates its recent performance over P by computing an average performance metric $\bar{P}$. This metric is compared against previous performance to detect potential concept drifts. If $\bar{P}$ has dropped by a threshold $\delta_{decrease}$, this indicates that the classifier may be underperforming, possibly due to outdated information in LTM. In response, the algorithm decreases $\lambda_{LTM}$, thereby reducing the size of LTM and giving STM a larger proportion of memory resources. This adjustment helps the classifier to focus more on recent data (stored in STM), which are likely more relevant to the current data distribution. After decreasing $\lambda_{LTM}$, LTM may be compressed, meaning that the oldest samples in LTM are removed to fit within the new, reduced memory allocation. Conversely, if $\bar{P}$ has improved by a threshold $\delta_{increase}$, this suggests that retaining older data might be beneficial. In this case, the algorithm increases $\lambda_{LTM}$, enlarging LTM’s memory allocation to preserve more long-term information. This adjustment helps the classifier to leverage older patterns that have become relevant again. Once either adjustment is made, the cool-down counter τ is reset, ensuring that memory allocation changes are infrequent enough to allow for stable adaptation to data changes without oscillating due to short-term performance fluctuations.

For each incoming unlabeled sample, the classifier predicts a label by choosing neighbors from either STM only or from both STM and LTM, depending on the specific configuration. If the classifier is configured to use only STM, it selects the nearest neighbors based on recent data alone, which might be suitable in highly volatile environments where only recent information is relevant. When using both STM and LTM, the classifier computes distances for *x* across both memory structures, selects the top *k* neighbors, and then uses either majority voting or distance-weighted voting to make a final prediction. This hybrid approach allows the classifier to balance recent trends with historical patterns when making decisions.

At regular intervals, the classifier updates the sizes of STM and LTM based on the current value of $\lambda_{LTM}$. The size of LTM is set to $\lambda_{LTM}\times M_{max}$, where $M_{max}$ is the total available memory. STM’s size $M_{STM}$ is then calculated as the remaining memory ($M_{max}-M_{LTM}$).

When LTM’s size exceeds its allocated memory $M_{LTM}$, the algorithm compresses LTM by removing the oldest samples until its size fits within $M_{LTM}$. This ensures that the classifier maintains memory constraints and prioritizes retaining more recent, potentially more relevant data within LTM.

The pseudocode of the proposed proposed classifier is shown in Algorithm 1.
**Algorithm 1:** proposed classifier**Input**: New sample x, label *y* (if available), hyperparameters**Output**: Predicted label $\hat{y}$ for x**Initialization:**
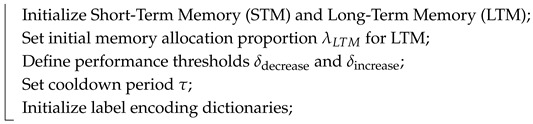
**For each incoming sample x:**
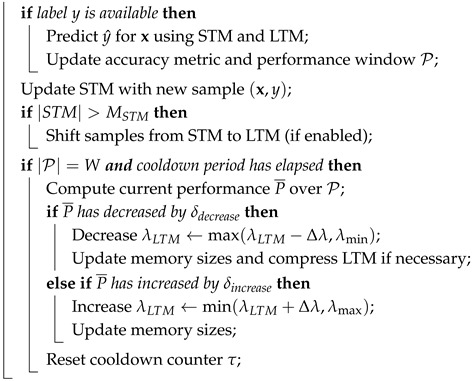
**Prediction Function**
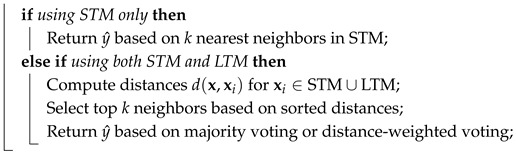
**UpdateMemorySizes()**
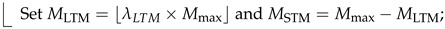
**CompressLTM()** 
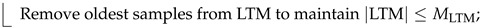


### 3.7. Complexity Analysis of the Adaptive SAMKNN

Understanding the computational complexity of Adaptive SAMKNN is crucial for assessing its scalability and efficiency in practical applications.

#### 3.7.1. Time Complexity

During training, the algorithm processes each incoming sample through several steps:Label Encoding: The algorithm assigns a unique numerical index to each new label encountered. This operation uses a dictionary lookup and insertion, performed in constant time, O(1), per sample.The new sample is appended to the STM, implemented as a deque. Appending to a deque is an O(1) operation.If the STM exceeds its maximum size, the oldest samples are shifted to the LTM. This involves the following:Deque Operations: Removing samples from the STM, which takes O(k), where k is the number of samples to shift.Updating LTM: Adding the shifted samples to the LTM. This may involve concatenating NumPy arrays, which have a time complexity of O(n + k), where n is the current size of the LTM.If the LTM exceeds its maximum size, the oldest samples are removed to maintain the size constraint. This involves slicing NumPy arrays, which is an O(1) operation because slicing creates a view rather than copying data.The algorithm updates a performance window (a deque) with the latest prediction result. This operation is O(1).It calculates the mean performance over the window, which is O(w), where w is the size of the performance window. Since w is a fixed parameter, this operation is effectively O(1) per sample.Based on performance evaluation, the algorithm may adjust the sizes of STM and LTM. This decision-making process is O(1).

The per-sample training time is dominated by operations that are either constant time or depend on fixed-size parameters. Therefore, the overall time complexity for training per sample is effectively O(1), assuming that the sizes of STM, LTM, and the performance window are bounded and relatively small.

Prediction involves several steps that are more computationally intensive:Calculating distances between the query sample and all samples in STM. This operation is O(s · d), where s is the number of samples in STM and d is the number of features.Similarly, calculating distances to all samples in LTM. This is O(l · d), where l is the number of samples in LTM.The total time for computing distances to both STM and LTM samples is O((s + l) · d).Instead of sorting all distances, the algorithm needs to find the top k nearest neighbors.Using a partial sort or a min-heap, this can be achieved in O((s + l) · log k).Aggregating the labels of the nearest neighbors to make a prediction. This operation is O(k).

The total time complexity for prediction per sample is
$$O\left((s+l)\cdot d+(s+l)\cdot \log k+k\right)$$

Since *k* is typically small and logk is negligible compared to *s* and *l*, the dominant term becomes
$$O\left((s+l)\cdot d\right)$$

This indicates that the prediction time scales linearly with the total number of samples in STM and LTM and the number of features.

#### 3.7.2. Space Complexity

The space required by the algorithm is determined by the storage of samples, labels, and auxiliary data structures:STM Storage: O(s · d), where s is the maximum size of STM. The space complexity of the STM storing a label is O(s), storing one label per sample.LTM Storage: O(l · d), where l is the maximum size of LTM. The space complexity of the LTM storing a label is O(l).Performance Window: O(w), where w is the size of the performance window.Label-to-Index Mapping: O(c), where c is the number of unique classes.

The total space complexity is
$$O\left((s+l)\cdot d+s+l+w+c\right)$$

Given that s and l are bounded by user-defined parameters (‘maxSTMSize’ and ‘maxLTMSize’), and w and c are relatively small, the space complexity is effectively linear in the maximum allowed memory sizes and the number of features:$$O\left(\mathrm{maxWindowSize}\cdot d\right)$$

## 4. Experimental Design

### 4.1. Experimental Setup

The experimental validation of the model developed in this work is presented in this subsection. Python 3.10 was utilized throughout the experimental validation to develop the proposed system using the River online ML framework [37]. We chose the River online ML framework because it can process streaming data. We used a MacBook Pro with an M1 chip running on 16 GB RAM and 1 TB disk storage to model the proposed system.

### 4.2. Datasets

We used three datasets to build and test our proposed IDS. The datasets are NF-BoT-IoT, NF-ToN-IoT, and NF-CSE-CIC-IDS 2018 proposed by [38]. The NetFlow records of these datasets were generated from the publicly available pcap files of the respective datasets. Table 3 shows the summary of the various datasets as proposed by [38]. We chose two attack classes from each dataset.

### 4.3. Evaluation Metrics

To comprehensively assess the effectiveness and efficiency of the proposed Adaptive SAMKNN system for zero-day attack detection in IoT and IIoT environments, a diverse set of evaluation metrics were employed. These metrics provide insights into various aspects of the system’s performance, including accuracy, reliability, computational efficiency, and statistical significance. The following sections define each metric and elucidate their relevance to evaluating the proposed intrusion detection system.

### 4.4. Accuracy

Accuracy measures the proportion of correctly classified instances (both true positives and true negatives) out of the total number of instances evaluated. It is calculated as
$$\mathrm{Accuracy}=\frac{\mathrm{True}\;\mathrm{Positives}+\mathrm{True}\;\mathrm{Negatives}}{\mathrm{Total}\;\mathrm{Instances}}$$

### 4.5. F1 Score

The F1 score is the harmonic mean of precision and recall, defined as
$$F1\;\mathrm{Score}=2\times \frac{\mathrm{Precision}\times \mathrm{Recall}}{\mathrm{Precision}+\mathrm{Recall}}$$
where precision is the ratio of true positives to the sum of true positives and false positives and recall (or detection rate) is the ratio of true positives to the sum of true positives and false negatives.

### 4.6. False Positive Rate (FPR)

FPR quantifies the proportion of benign instances incorrectly classified as malicious. It is calculated as
$$\mathrm{FPR}=\frac{\mathrm{False}\;\mathrm{Positives}}{\mathrm{False}\;\mathrm{Positives}+\mathrm{True}\;\mathrm{Negatives}}$$

### 4.7. Detection Rate (DR)

Detection rate, also known as recall or sensitivity, measures the proportion of actual malicious instances correctly identified by the system. It is expressed as
$$\mathrm{Detection}\;\mathrm{Rate}=\frac{\mathrm{True}\;\mathrm{Positives}}{\mathrm{True}\;\mathrm{Positives}+\mathrm{False}\;\mathrm{Negatives}}$$

DR is fundamental in evaluating the system’s effectiveness in identifying real threats. A high detection rate indicates the system’s proficiency in recognizing and mitigating malicious activities, which is paramount for maintaining the security integrity of IoT and IIoT networks.

### 4.8. Processing Time

Processing time refers to the duration required by the system to analyze and classify each data instance. It is typically measured in milliseconds or seconds per instance.

In real-time intrusion detection, rapid processing is essential for ensuring timely responses to threats. Monitoring processing time assesses the system’s capability to operate efficiently under high-throughput conditions typical of IoT environments.

### 4.9. Paired T-Test

The paired *t*-test is a statistical method used to compare the means of two related groups to determine if there is a significant difference between them.

The formula for the paired *t*-test:$$t=\frac{\bar{d}}{s_{d}/\sqrt{n}}$$
where

$\bar{d}$ is the mean of the differences, $s_{d}$ is the standard deviation of the differences, and *n* is the number of paired observations.

Applying the paired *t*-test allows for the evaluation of the statistical significance of performance improvements achieved by the proposed classifier over baseline classifiers. It ensures that observed differences in metrics like accuracy or F1 score are not due to random chance.

### 4.10. Brier Score

The Brier score measures the accuracy of probabilistic predictions by calculating the mean squared difference between predicted probabilities and the actual binary outcomes. It ranges from 0 to 1, with lower scores indicating better calibration.
$$\mathrm{Brier}\;\mathrm{Score}=\frac{1}{N}\sum_{i=1}^{N}{\left(f_{i}-o_{i}\right)}^{2}$$
where $f_{i}$ is the predicted probability and $o_{i}$ is the actual outcome.

The Brier score evaluates the reliability of the probability estimates produced by the classifier, ensuring that the system not only makes accurate predictions but also provides well-calibrated confidence levels in its detections.

### 4.11. Bootstrap Confidence Interval

Bootstrap confidence intervals are statistical intervals estimated by repeatedly resampling the dataset with replacement and computing the metric of interest across these samples to determine the range within which the true metric value lies with a certain confidence level.

Bootstrap confidence intervals provide insights into the variability and reliability of the performance metrics, enabling a more robust understanding of the system’s performance across different data samples.

### 4.12. CPU Usage

CPU usage measures the percentage of central processing unit resources consumed by the system during operation.

Monitoring CPU usage is essential for evaluating the computational efficiency of the proposed classifier, especially in resource-constrained IoT devices where excessive CPU consumption can lead to performance bottlenecks and increased energy consumption.

### 4.13. Memory Usage

Memory Usage quantifies the amount of RAM utilized by the system during its operation.

Efficient memory usage is critical for deploying intrusion detection systems on IoT devices with limited memory resources. Assessing memory consumption ensures that Adaptive SAMKNN can operate effectively without exceeding the device’s memory constraints, thereby facilitating broader deployment in diverse IoT environments.

The selected metrics collectively provide a holistic evaluation of the proposed classifier’s performance in detecting zero-day attacks within IoT and IIoT systems. Accuracy and F1 score offer general performance insights, while FPR and DR specifically address the balance between detecting true threats and minimizing false alarms, which is crucial for maintaining system reliability. Processing time, CPU usage, and memory usage assess the system’s operational efficiency, ensuring its practicality for real-time applications in resource-constrained environments. Statistical measures like the paired *t*-test, t-statistic, Brier score, and bootstrap confidence interval validate the robustness and reliability of the performance improvements, ensuring that the system’s enhancements are both significant and consistent. Collectively, these metrics ensure that the proposed classifier not only excels in identifying novel and emerging threats but also operates efficiently within the stringent resource limitations typical of IoT and IIoT deployments.

### 4.14. Experimental Validations

Eight experiments were conducted to test and validate our proposed system. The experiments are briefly described below.

#### 4.14.1. Baseline Performance Evaluation

The first experiment evaluated the performance of the proposed classifier on the three datasets while recording the accuracy, F1 score, processing time, and memory usage. As part of the baseline performance evaluation, we looked at the performance of five popular offline ML algorithms, namely, random forest (RF), KNN, support vector machines (SVMs), decision tree (DT), and logistic regression (LR), and compared their performance to our proposed technique. We also compared the performance of our proposed method with the original SAMKNN algorithm.

#### 4.14.2. Zero-Day Attack Detection

This experiment aims to evaluate the effectiveness of the proposed classifier in detecting both known and zero-day attacks within IoT network traffic data. Zero-day attacks are novel threats that have not been previously encountered or included in the training data, posing significant challenges for intrusion detection systems. By simulating zero-day attacks and assessing the classifier’s performance, we seek to understand its capability in real-world scenarios where new types of attacks emerge. Two different experiments were carried out under this experiment. During the first experiment, we used some attack classes from the datasets used to validate our system as zero-day attacks. In the second experiment, we used the conditional tabular generative adversarial network (CTGAN) proposed by [39] to generate synthetic zero-day attacks into the original datasets. The CTGAN is a specialized type of GAN designed to generate realistic synthetic tabular data by conditioning on specific data features. It consists of two main components: a generator that creates fake data samples and a discriminator that evaluates whether samples are real or synthetic, both trained simultaneously to improve the quality of generated data. Additionally, it employs a training strategy that focuses on capturing the relationships and dependencies between various features in the dataset, enhancing the fidelity of the synthetic data. The zero-day attacks were simulated by excluding some attack instances from the training dataset, ensuring that the classifier had not been exposed to this attack type during training. These unseen instances were then introduced into the testing dataset as zero-day attacks, allowing the evaluation of the model’s ability to detect novel and previously unencountered threats. In the NF-BoT IoT dataset, Reconnaissance, DDoS, and Theft were used to simulate zero-day attacks. Injection, Malware, and DDoS from the NF-ToN IoT and NF-CSE datasets were used to simulate zero-day attacks. A crucial metric for practical applications is indicating how often benign traffic is incorrectly classified as malicious. The average time taken to predict each instance, indicating the model’s efficiency, was also measured.

#### 4.14.3. Scalability

In this experiment, we evaluated the scalability and performance of the Adaptive SAMKNN on a large-scale IoT network traffic dataset. The primary focus was to assess how the classifier performs in terms of predictive accuracy and resource utilization (CPU and memory usage) when processing data in a streaming, online learning context. By monitoring these aspects, we aimed to determine the model’s suitability for real-time intrusion detection in IoT environments, where accuracy and efficiency are critical. We measured these metrics over samples of 50,000, 200,000, 350,000, and 500,000.

#### 4.14.4. False Positive Rate Evaluation Under Normal Conditions

This experiment assessed the performance of the proposed technique in terms of its false positive rate when exposed solely to benign network traffic. Understanding the FPR under normal operating conditions is crucial for IDSs, as a high rate of false alarms can lead to alert fatigue, resource wastage, and decreased trust in the system. A low false positive rate ensures that security personnel can focus on genuine threats without being overwhelmed by false alarms.

#### 4.14.5. Performance Under Drift

In the rapidly evolving landscape of network security, particularly within the IoT and IIoT ecosystem, detecting malicious activities is a complex and dynamic challenge. IDSs must not only identify known threats but also adapt to changes in network behavior over time. One significant challenge in this domain is concept drift, where the statistical properties of the target variable change over time in unforeseen ways. This experiment evaluated the performance of our proposed Adaptive SAMKNN under various types of drift scenarios simulated in IoT network traffic data. This experiment considered four types of drifts: gradual, sudden, recurring, and incremental. Our simulated concept drift was executed by introducing variations in the class label of the dataset.

#### 4.14.6. Resource Utilization

In this experiment, we aimed to compare the resource utilization and processing efficiency of the proposed classifier with the standard SAMKNN classifier when applied to IoT network traffic data. By evaluating metrics such as CPU usage, memory consumption, and processing time, we sought to understand the computational overhead introduced by the adaptive capabilities of the proposed classifier. This comparison is crucial for determining the practicality of deploying adaptive models in resource-constrained environments typical of IoT networks. To obtain a good approximation of the amount of resources that the proposed model consumes, we ran ten experiments while recording the resource usage and then calculated the mean of those experiments.

#### 4.14.7. Statistical Analysis

To assess the proposed model’s robustness and ensure that the results are replicable and not randomized, we performed statistical analysis using the bootstrap confidence interval, Brier score, paired *t*-test, and t-statistic.

The bootstrap confidence interval assesses the performance of the proposed classifier in detecting zero-day attacks within IoT networks by applying 100 bootstrap samples to each dataset and evaluating the accuracy of each iteration. The resulting accuracies are used to calculate the mean and 95% confidence interval, which are then visualized with a histogram to demonstrate the classifier’s reliability and consistency.

Another statistical analysis involves assessing the calibration accuracy of the proposed classifier on the respective dataset by implementing a custom Brier score metric to evaluate the precision of predicted probabilities for the positive class. The algorithm iterates over each instance to obtain probability predictions, and, updating the Brier score accordingly, this study quantifies the reliability of the classifier’s probabilistic outputs in effectively detecting zero-day attacks in IoT environments.

On each dataset, we compared the performance of the proposed classifier and original SAMKNN by conducting five independent runs with randomly shuffled data samples, measuring each classifier’s accuracy in detecting zero-day attacks. The resulting accuracy scores were statistically analyzed using a paired *t*-test to determine the significance of the performance differences between the two classifiers.

#### 4.14.8. Ablation Study

In machine learning, an ablation study is a crucial experimental approach used to assess the contribution of individual components or features within a model. By systematically removing or altering these components, researchers can determine their impact on the overall performance of the system. This experiment involves conducting an ablation study on the proposed classifier within the context of IoT network intrusion detection. The goal is to understand how different aspects of the classifier influence its ability to detect malicious activities in network traffic data. The components removed are the adaptiveness, LTM, and SAM.

## 5. Results

### 5.1. Baseline Performance Evaluation

The experimental results presented in Table 4, Table 5, Table 6 and Table 7 demonstrate the performance of the SAMKNN, Adaptive SAMKNN, and five offline ML algorithms across three benchmark datasets: NF-BoT-IoT, NF-ToN-IoT, and NF-CSE-CIC-IDS 2018. The Adaptive SAMKNN consistently outperforms the SAMKNN in memory efficiency across all datasets, achieving substantial reductions in memory usage while maintaining comparable accuracy and F1 scores. Specifically, Adaptive SAMKNN achieves memory savings of over 99% compared to SAMKNN without compromising detection accuracy, demonstrating its suitability for resource-constrained environments. Additionally, compared with traditional machine learning algorithms such as RF, KNN, SVM, DT, and LR, the Adaptive SAMKNN shows competitive accuracy and a low processing time, making it a viable option for real-time attack detection in IoT environments. These results highlight the effectiveness of Adaptive SAMKNN in balancing accuracy, memory efficiency, and processing speed.

### 5.2. Zero-Day Attack Detection

Table 8 and Table 9 present the performance of the proposed classifier in detecting zero-day attacks across three datasets (NF-BoT-IoT, NF-ToN-IoT, and NF-CSE-CIC-IDS 2018). The Adaptive SAMKNN demonstrates robust accuracy, achieving over 99% across all attack classes, with a consistent zero-day detection rate (DR) of 1.00, indicating that it successfully identified all simulated zero-day attacks. The classifier maintains a low FPR, particularly for NF-ToN-IoT and NF-CSE-CIC-IDS 2018, while the NF-BoT-IoT dataset shows slightly higher FPR values, especially for Theft and DDoS attacks. Additionally, the average detection latency remains minimal at 0.0001 s across all datasets, supporting its suitability for real-time applications. These results illustrate that the Adaptive SAMKNN can effectively identify and respond to novel threats with high precision and minimal delay, making it a promising solution for dynamic IoT and IIoT security challenges.

### 5.3. Scalability

The scalability assessment of Adaptive SAMKNN, as shown in Figure 2, Figure 3, Figure 4 and Table 10, demonstrates its capability to maintain consistent performance across increasing data volumes on three datasets: NF-BoT-IoT, NF-ToN-IoT, and NF-CSE-CIC-IDS 2018. With sample sizes growing from 50,000 to 500,000, Adaptive SAMKNN achieves stable accuracy, retaining over 98% accuracy on NF-BoT-IoT, 96% on NF-ToN-IoT, and 99% on NF-CSE-CIC-IDS 2018. CPU usage remains minimal (around 12.4%) across all datasets, and memory usage is kept consistently low, especially on NF-BoT-IoT with 45.84 MB, and slightly higher on the other datasets. Processing times increase proportionally with sample size but remain manageable for large datasets. The scalability test confirms that Adaptive SAMKNN maintains high accuracy, low CPU usage, and efficient memory management, underscoring its effectiveness for real-time applications even as data volume scales.

### 5.4. False Positive Rate Evaluation Under Normal Conditions

Table 11 shows the false positive rate (FPR) of the proposed classifier under normal conditions across three datasets: NF-BoT-IoT, NF-ToN-IoT, and NF-CSE-CIC-IDS 2018. The results indicate that Adaptive SAMKNN achieved an FPR of 0.00% on all datasets, demonstrating its ability to identify normal traffic without false alarms correctly. With 13,858, 179,646, and 62,466 normal samples in the NF-BoT-IoT, NF-ToN-IoT, and NF-CSE-CIC-IDS 2018 datasets, respectively, the classifier consistently maintained perfect FPR performance. These findings underscore Adaptive SAMKNN’s precision in distinguishing benign and malicious activities under stable, non-attack conditions. The absence of false positives is a significant outcome, as it highlights the model’s reliability in recognizing genuine network behavior, minimizing the risk of alert fatigue in real-world monitoring environments where excessive false positives can hinder security operations.

### 5.5. Performance Under Drift

Table 12, Table 13 and Table 14 present the performance of Adaptive SAMKNN in handling different types of concept drift across three datasets: NF-BoT-IoT, NF-ToN-IoT, and NF-CSE-CIC-IDS 2018. The classifier exhibits varied accuracy and F1 scores based on the nature of the drift. Adaptive SAMKNN achieves the highest performance for gradual drift, with accuracy and F1 scores exceeding 90% across all datasets. Sudden drift poses a greater challenge, resulting in the lowest performance, particularly on the NF-ToN-IoT dataset, where accuracy and F1 scores drop to around 81%. Performance under recurring and incremental drift remains stable, with accuracy and F1 scores generally above 87%. These results indicate that while Adaptive SAMKNN is capable of adapting to gradual changes in data distribution, its sensitivity to sudden changes suggests potential areas for improvement in environments with abrupt shifts. Overall, the classifier demonstrates robust adaptability to diverse drift types, which is essential for real-world IoT applications with dynamic data patterns.

### 5.6. Resource Utilization

Table 15 presents a comparative analysis of resource utilization between SAMKNN and Adaptive SAMKNN across the NF-BoT-IoT, NF-ToN-IoT, and NF-CSE-CIC-IDS 2018 datasets. The Adaptive SAMKNN consistently demonstrates a lower mean CPU usage, memory consumption, and processing time compared to the standard SAMKNN. For instance, in the NF-BoT-IoT dataset, Adaptive SAMKNN achieves a mean CPU usage of 86.51% and memory consumption of 3.20 KB, whereas SAMKNN requires 94.51% CPU usage and 76.80 KB memory. Across all datasets, Adaptive SAMKNN maintains a processing time of 0.07 s, slightly faster than SAMKNN’s 0.10 s. These findings emphasize Adaptive SAMKNN’s computational and memory resource efficiency, making it more suitable for real-time applications in resource-constrained environments like IoT networks. The reduced resource demands of Adaptive SAMKNN, without sacrificing detection accuracy, underscore its advantage for deployment in practical, high-frequency data processing scenarios.

### 5.7. Statistical Analysis

The bootstrap confidence intervals for the proposed classifier demonstrate high accuracy across the three datasets, as shown in Figure 5, Figure 6 and Figure 7, with narrow ranges indicating consistent performance. For the NF-BoT-IoT dataset, the classifier achieves a mean accuracy of 0.9813, with a 95% confidence interval between 0.9800 and 0.9826. Similarly, on the NF-ToN-IoT dataset, the mean accuracy is 0.9635, with a 95% confidence interval of 0.9612 to 0.9658. The NF-CSE-2018 dataset shows the highest accuracy, with a mean of 0.9968 and a 95% confidence interval from 0.9962 to 0.9972.

Additionally, the Brier scores indicate strong calibration, as shown in Table 16, with values of 0.0166, 0.0260, and 0.0028 for the NF-BoT-IoT, NF-ToN-IoT, and NF-CSE-2018 datasets, respectively.

The *t*-tests, as shown in Table 17, revealed significant differences between the proposed classifier and SAMKNN, with *p*-values below 0.001 across all datasets.

### 5.8. Ablation Study

The ablation study results in Table 18 compare the performance of Adaptive SAMKNN and SAMKNN with and without long-term memory (LTM) across three datasets: NF-BoT-IoT, NF-ToN-IoT, and NF-CSE-CIC-IDS 2018. Adaptive SAMKNN without LTM consistently achieves high accuracy and F1 scores, comparable to the SAMKNN configurations, while maintaining minimal memory usage (approximately 73.91 KB). Including LTM in SAMKNN leads to a substantial increase in memory consumption, with up to 59,002.88 KB for NF-ToN-IoT, highlighting the efficiency of Adaptive SAMKNN without LTM for memory-constrained environments. Moreover, Adaptive SAMKNN and SAMKNN without LTM demonstrate faster processing times than KNN, significantly reducing the computational load. These findings emphasize Adaptive SAMKNN’s capability to provide a balanced trade-off between accuracy, processing speed, and memory efficiency, making it suitable for real-time IoT applications where computational resources are often limited.

## 6. Discussions

The findings highlight the proposed classifier as a viable solution for real-time anomaly detection in IoT networks, effectively addressing critical constraints in resource-limited environments. Unlike the traditional SAMKNN, which demands substantial memory resources, Adaptive SAMKNN significantly reduces memory consumption while maintaining high accuracy. This efficiency enhances its suitability for dynamic, high-throughput data streams typical in IoT systems. Comparative analyses with standard classifiers reveal Adaptive SAMKNN’s competitive advantage, especially in scenarios that require rapid, reliable decision making with minimal computational overhead. These performance attributes position Adaptive SAMKNN as a key component for scalable and sustainable IoT security frameworks. Future research may focus on its application to larger, more heterogeneous datasets and its integration with other adaptive learning frameworks to bolster resilience against evolving cyber threats within IoT ecosystems.

Experimental results demonstrate Adaptive SAMKNN’s proficiency in detecting zero-day attacks with exceptional accuracy and low false positive rates (FPRs) across various IoT datasets. Notably, the classifier achieved a perfect detection rate of 1.00 for zero-day attacks, underscoring its robustness in identifying previously unseen threats. Although the NF-BoT-IoT dataset showed a slightly higher FPR, the rates remained within manageable limits, indicating that minor model adjustments could further enhance performance. The classifier’s minimal detection latency underscores its applicability in real-time security applications, where swift responses are essential. Adaptive SAMKNN’s adaptive nature ensures sustained high accuracy and efficiency compared to traditional classifiers that may falter with evolving threats. Future studies could aim to refine the model to reduce the FPR across diverse datasets further and improve its adaptability in more complex IoT environments.

Scalability analysis reveals Adaptive SAMKNN’s capability to manage large-scale data effectively, making it ideal for IoT environments with continuously growing data streams. The classifier maintains high accuracy despite increasing sample sizes, demonstrating stability and reliability in monitoring extensive datasets without performance degradation. Its low CPU and memory usage emphasize its efficiency, which is crucial for resource-constrained IoT settings where excessive resource consumption can impede practical deployment. Although processing time gradually increases with larger datasets, Adaptive SAMKNN remains responsive within acceptable limits, preserving its real-time applicability. These results confirm Adaptive SAMKNN’s ability to scale efficiently, adapting to high-throughput data while maintaining accuracy and resource efficiency. Future work could explore further optimizations in processing speed to handle even larger data volumes, enhancing its applicability across broader IoT and cybersecurity applications.

Under normal conditions, Adaptive SAMKNN exhibits a zero false positive rate (FPR) across all tested datasets, indicating exceptional reliability in distinguishing legitimate network activity without generating unnecessary alerts. This zero FPR is particularly valuable in practical deployments, where false positives can disrupt operations and cause alert fatigue among security personnel. The classifier’s precision in differentiating benign traffic from potential threats underscores its effectiveness for real-time IoT deployments, where accurate monitoring is critical for maintaining network integrity. These results suggest that Adaptive SAMKNN can be reliably deployed in production environments with minimal risk of false alarms, enhancing efficient and focused anomaly detection. Future research could investigate its resilience under dynamic conditions to ensure a sustained low FPR in varying operational contexts.

Adaptive SAMKNN demonstrates robust performance under various drift conditions, effectively handling gradual, recurring, and incremental drifts, common in IoT data patterns. This adaptability aligns well with the evolving nature of IoT environments. However, the classifier experiences performance declines in sudden drift scenarios, highlighting a need for enhanced mechanisms such as improved memory adjustment or rapid drift detection to better respond to abrupt changes. While Adaptive SAMKNN is well suited for stable and gradually changing IoT environments, further research is necessary to bolster its robustness against sudden drifts, ensuring broader applicability across diverse IoT security contexts.

Resource utilization comparisons indicate that Adaptive SAMKNN is more efficient than traditional SAMKNN, making it well suited for IoT environments with limited computational and memory resources. Its lower CPU and memory usage across all datasets enable effective real-time operation without overburdening system resources, which is critical for IoT network deployments. Its reduced processing time supports its adaptability to high-speed data streams, ensuring rapid anomaly detection and response. These efficiency gains align with the goals of IoT security, which prioritize minimizing resource consumption to enable continuous monitoring and analysis. Adaptive SAMKNN’s demonstrated efficiency suggests its practicality for real-time anomaly detection in IoT applications. Future investigations could assess its scalability in even more constrained environments and explore its integration into lightweight IoT frameworks for broader deployment.

The statistical analysis underscores the robustness and reliability of the proposed classifier, as evidenced by its high accuracy and tight confidence intervals across diverse IoT datasets. The low Brier scores reflect the model’s strong predictive calibration, which is particularly critical for IoT applications that demand reliability. The statistically significant *t*-tests suggest that the classifier consistently outperforms the SAMKNN baseline, highlighting its superiority in real-world scenarios. Furthermore, the performance on the NF-CSE-2018 dataset, with its near-perfect accuracy, suggests that the classifier is highly effective in scenarios with minimal noise or well-structured data. These findings collectively emphasize the potential of the proposed approach for accurate and reliable IoT network traffic classification.

The ablation study emphasizes the role of long-term memory (LTM) in SAMKNN and the proposed classifier, particularly regarding memory usage and processing efficiency. While LTM enhances SAMKNN’s adaptability, it introduces significant memory overhead, making it less feasible for resource-limited IoT environments. In contrast, Adaptive SAMKNN without LTM offers a more memory-efficient solution without sacrificing accuracy, making it preferable for dynamic IoT networks. Additionally, Adaptive SAMKNN shows a substantial reduction in processing time compared to the KNN baseline, reinforcing its suitability for real-time applications that require swift decision making. These results indicate that Adaptive SAMKNN without LTM strikes an optimal balance for IoT anomaly detection, providing reliable performance while minimizing resource demands.

### 6.1. Limitations

Despite its promising performance, the proposed technique exhibits several limitations that warrant attention. Firstly, the classifier is sensitive to sudden drifts in data distributions, leading to temporary declines in detection accuracy during abrupt changes. This sensitivity underscores the necessity for more robust drift detection mechanisms to enhance responsiveness to rapid shifts in network behavior. Secondly, the proposed classifier’s effectiveness is somewhat dependent on the specific datasets used for training and evaluation. The model may not generalize uniformly across all types of IoT and IIoT datasets, particularly those with unique features, potentially limiting its applicability in diverse real-world scenarios. Additionally, while the proposed algorithm is designed to be resource-efficient, it still requires a baseline level of computational and memory resources that may be challenging for some extremely low-resource-constrained IoT devices. This constraint can limit the deployment of the classifier on lower-end devices or in environments where resources are highly limited. Addressing these limitations through targeted enhancements, such as incorporating advanced drift detection algorithms and optimizing resource utilization, is essential for improving the robustness and versatility of the proposed algorithm in practical IoT deployments.

### 6.2. Real-World Deployment Scenarios and Challenges

Deploying the proposed algorithm in real-world IoT and IIoT environments presents a multitude of promising opportunities alongside significant challenges. In practical scenarios, the proposed algorithm can be integrated into diverse applications such as smart grids, industrial automation systems, healthcare monitoring networks, and smart home infrastructures. For instance, in smart grids, the model can continuously monitor network traffic to detect and mitigate cyber threats that could disrupt power distribution and critical services. Similarly, in industrial automation, the model can protect automated machinery and control systems from malicious intrusions that could lead to operational downtimes or safety hazards.

However, the deployment of the proposed classifier is not without its challenges. One of the primary obstacles is the integration with existing legacy systems, which may have limited compatibility with modern machine learning frameworks. Additionally, the resource-constrained nature of many IoT devices poses another challenge, as deploying even lightweight models can strain limited computational and memory resources. Although the proposed classifier is designed for efficiency, optimizing its deployment to operate within the stringent constraints of edge devices necessitates further refinement.

Real-time processing requirements add another layer of complexity, as the model must deliver prompt detection and response to threats without introducing latency that could disrupt critical operations. Achieving this necessitates robust optimization techniques and possibly distributed processing architectures to balance the load effectively.

Scalability is also a critical factor, as real-world IoT deployments often involve vast and continuously growing networks of devices. The proposed algorithm must maintain its performance and accuracy across expanding datasets and increasingly complex network topologies.

## 7. Conclusions

This study demonstrates that Adaptive SAMKNN is a highly effective and efficient solution for anomaly detection in IoT environments. Through extensive evaluations, Adaptive SAMKNN proves its robustness in handling various datasets, including NF-BoT-IoT, NF-ToN-IoT, and NF-CSE-CIC-IDS 2018, achieving high accuracy, low memory usage, and minimal CPU consumption. The model’s consistent performance across these datasets, even with large data volumes, underscores its scalability, making it a suitable candidate for real-time applications in resource-constrained IoT networks.

One of the key strengths of Adaptive SAMKNN is its ability to maintain a zero false positive rate under normal conditions, a critical factor in ensuring the reliability of IoT security frameworks. This capability reduces alert fatigue, enabling security teams to focus on genuine threats without being overwhelmed by false alarms. The model’s low false positive rate and high detection rate make it a dependable choice for distinguishing between benign and malicious network activities, even in dynamic and evolving environments.

This study also explores the impact of different types of concept drift, revealing that Adaptive SAMKNN effectively adapts to gradual, recurring, and incremental drifts. However, it encounters challenges in sudden drift scenarios, where performance slightly declines. This limitation suggests an opportunity for further improvements, such as incorporating rapid drift detection mechanisms to enhance responsiveness in highly volatile environments.

Ablation studies further reveal that Adaptive SAMKNN’s performance remains robust even without long-term memory, significantly reducing memory usage while maintaining accuracy and F1 scores. This makes the model highly efficient for real-time applications, where minimizing resource consumption is essential. Additionally, the scalability assessment confirms that Adaptive SAMKNN can handle increasing data volumes without sacrificing performance, highlighting its potential for broader IoT deployments.

Future research directions include improving the model’s adaptability to sudden drift and exploring its integration with other adaptive learning frameworks. To achieve this and integrate seamlessly with lightweight IoT devices, we can incorporate advanced drift detection algorithms like ADWIN and DDM into the Adaptive SAMKNN framework to swiftly identify abrupt data distribution changes. Additionally, we can further optimize the model for resource-constrained environments through techniques such as model pruning, quantization, and the use of lightweight data structures to reduce memory and processing requirements.

## Figures and Tables

**Figure 1 sensors-25-00216-f001:**
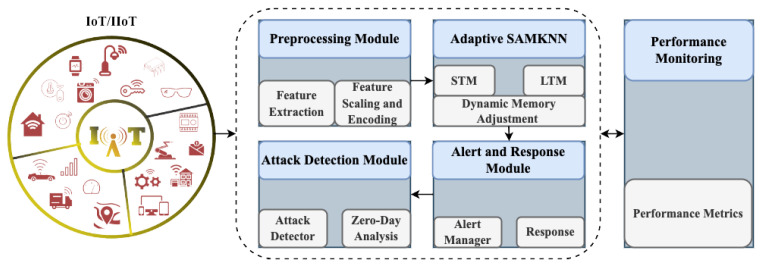
An architectural diagram of our proposed IDS.

**Figure 2 sensors-25-00216-f002:**
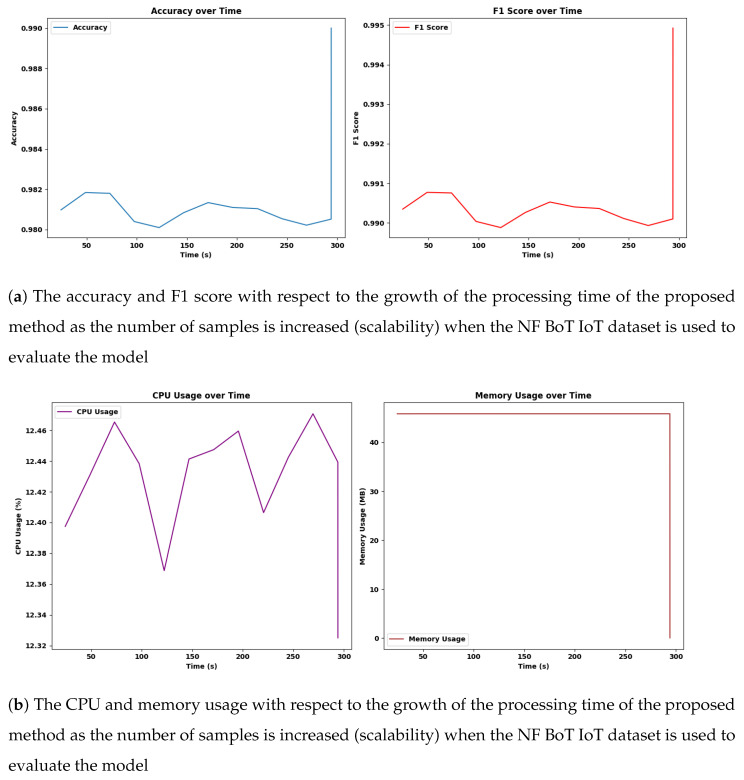
The accuracy, F1 score, CPU, and memory usage with respect to the growth of the processing time of the proposed method as the number of samples is increased (scalability) when the NF BoT IoT dataset is used to evaluate the model.

**Figure 3 sensors-25-00216-f003:**
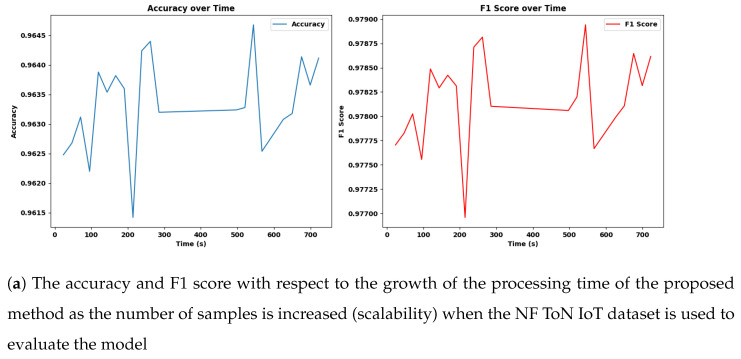
The accuracy, F1 score, CPU, and memory usage with respect to the growth of the processing time of the proposed method as the number of samples is increased (scalability) when the NF ToN IoT dataset is used to evaluate the model.

**Figure 4 sensors-25-00216-f004:**
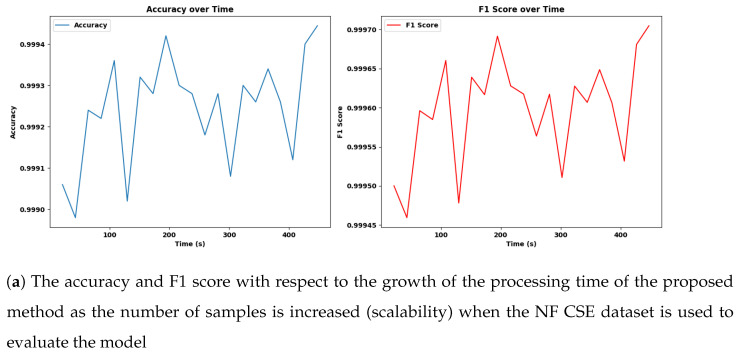
The accuracy, F1 score, CPU, and memory usage with respect to the growth of the processing time of the proposed method as the number of samples is increased (scalability) when the NF CSE dataset is used to evaluate the model.

**Figure 5 sensors-25-00216-f005:**
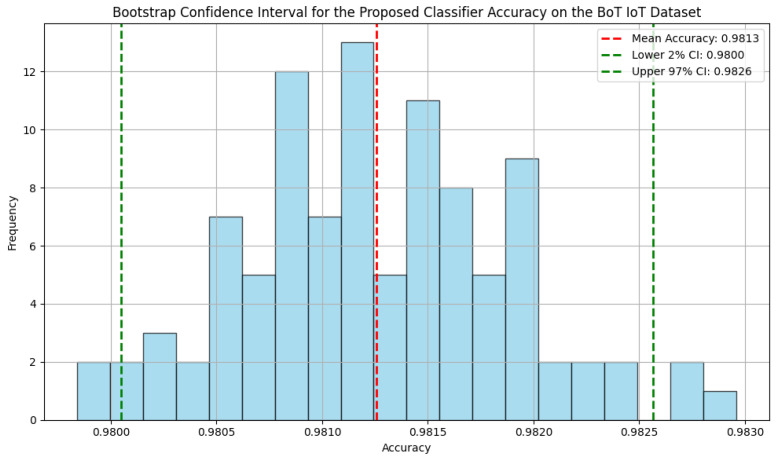
The bootstrap confidence interval for the proposed classifier accuracy on the NF BoT IoT dataset.

**Figure 6 sensors-25-00216-f006:**
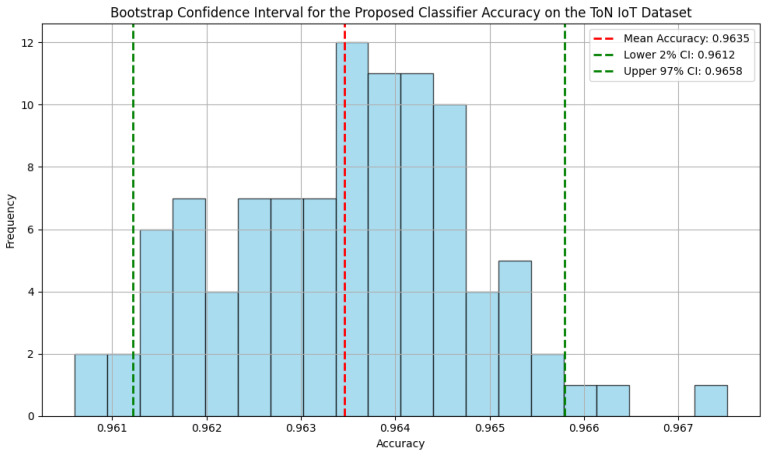
The bootstrap confidence interval for the proposed classifier accuracy on the NF ToN IoT dataset.

**Figure 7 sensors-25-00216-f007:**
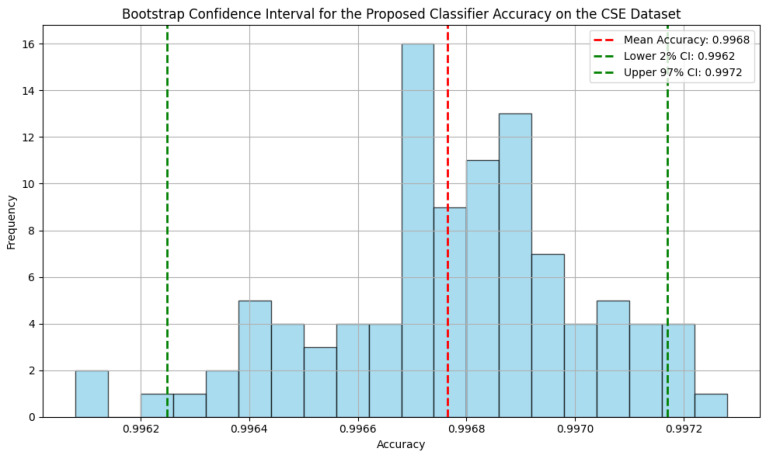
The bootstrap confidence interval for the proposed classifier accuracy on the NF CSE 2018 dataset.

**Table 1 sensors-25-00216-t001:** Comparison of existing methods and the proposed Adaptive SAMKNN for zero-day attack detection in IoT and IIoT environments.

Reference	Approach	Zero-Day Detection	Resource Requirements	Performance Metrics	Strengths	Limitations
[14]	Novel aggregation algorithm for federated learning to detect zero-day botnet attacks	Effective	Moderate computational resources	Superior model aggregation and detection performance	Enhances detection while preserving privacy, minimizes communication overhead	Complexity in handling heterogeneous data across different IoT devices
[15]	Utilizes CNN with regularization techniques for anomaly detection	Effective	Moderate to high computational resources	Improved detection performance with regularization	Enhanced detection capability with regularization	May not generalize well across heterogeneous IoT environments
[16]	Anomaly detection using CNN enhanced with regularization methods	Effective	Similar to Hairab et al., 2022	High detection accuracy for zero-day attacks	Improved robustness against overfitting and better generalization	High resource consumption makes it less suitable for resource-constrained devices
[17]	IDS at the data processing layer of the Internet of Vehicles, compared with SVM and RF	Effective	Moderate computational resources	Higher detection accuracy than SVM and RF	Faster detection rate by operating at the data processing layer	Performance may degrade on different device types due to CNN’s dependency on specific architectures
[31]	Uses online RL for moving target defense (MTD) with behavioral fingerprinting	Effective (except rootkits)	Low storage (1 MB) and 10% RAM usage	Successfully mitigates all tested attacks except rootkits	Low resource consumption, adaptable MTD strategies	Unable to mitigate rootkits, limited by RL model’s adaptability to certain attack types
[32]	IDS framework using TL based on CNN to detect known and zero-day attacks	Effective	High computational resources due to CNN and TL	High detection rates for both known and zero-day attacks	Leverages pre-trained models for improved detection capabilities	High resource demands make it unsuitable for many IoT devices

**Table 2 sensors-25-00216-t002:** Comparison of existing methods and the proposed Adaptive SAMKNN for zero-day attack detection in IoT and IIoT environments.

Reference	Approach	Zero-Day Detection	Resource Requirements	Performance Metrics	Strengths	Limitations
[22,23,24]	Combines ML classifiers, DNNs, and DRL for dynamic model selection	Highly Effective	Highly computationally intensive	99% detection rate with minimal errors	Robust defense mechanism with high detection rates and minimal false positives/negatives	Computationally intensive, challenging for deployment on resource-constrained IoT devices
[29]	Builds IDS using an ensemble of deep learning classifiers trained on benchmark datasets	Effective	High computational and memory usage	High performance in detecting zero-day attacks	Validated on multiple datasets, effective against unknown attacks	High resource consumption limits deployment on resource-constrained IoT devices
**Proposed Method**	**Adaptive k-nearest neighbors with self-adjusting memory (SAM) for anomaly detection**	**Highly Effective**	**Low CPU and memory consumption (as low as 0.05 MB)**	**High accuracy and F1 scores; Detection rate of 1.00 for zero-day attacks**	**Real-time, scalable, memory-efficient, maintains performance with large data volumes**	**Sensitive to sudden data drift, requiring further enhancement for handling abrupt changes**

**Table 3 sensors-25-00216-t003:** Summary of the datasets as proposed by [38].

Dataset	# Features	Benign	# Samples
NF-BoT-IoT	12	13,859 (2.31%)	600,100
NF-ToN-IoT	12	179,647 (13.94%)	1,288,642
NF-CSE-CIC-IDS 2018	12	7,373,198 (87.86%)	8,392,401

**Table 4 sensors-25-00216-t004:** Comparing the accuracy, F1 score, and model memory footprint (KB) of our proposed classifier against the original SAMKNN algorithm.

Algorithm	Accuracy	F1	Memory (KB)
NF-BoT-IoT			
SAMKNN	98.52%	98.50%	5539.84
Adaptive SAMKNN	98.90%	98.89%	62.51
NF-ToN-IoT			
SAMKNN	95.76%	95.61%	59,002.88
Adaptive SAMKNN	96.34%	96.23%	63.31
NF-CSE-CIC-IDS 2018			
SAMKNN	99.91%	99.91%	57,477.12
Adaptive SAMKNN	99.92%	99.92%	55.11

**Table 5 sensors-25-00216-t005:** Comparing the accuracy, processing time (s), and model memory footprint (MB) of our proposed classifier against five other popular ML algorithms when evaluated on NF BoT IoT dataset.

Algorithm	Accuracy	Time (s)	Memory (MB)
RF	98.93%	14.98	1436.41
KNN	99.13%	26.47	1439.48
SVM	98.81%	2550.37	2043.38
DT	98.96%	1.62	1366.13
LR	98.69%	2.70	982.86
Our method	98.90%	11	0.06

**Table 6 sensors-25-00216-t006:** Comparing the accuracy, processing time (s), and model memory footprint (MB) of our proposed classifier against five other popular ML algorithms when evaluated on NF ToN IoT dataset.

Algorithm	Accuracy	Time (s)	Memory (MB)
RF	99.93%	79.51	1152.52
KNN	99.34%	102.49	896.75
SVM	97.94%	10,364.42	2547.66
DT	999.91%	3.57	2101.47
LR	995.98%	2.54	2053.06
Our method	96.34%	187	0.06

**Table 7 sensors-25-00216-t007:** Comparing the accuracy, processing time (s), and model memory footprint (MB) of our proposed classifier against five other popular ML algorithms when evaluated on NF CSE 2018 dataset.

Algorithm	Accuracy	Time (s)	Memory (MB)
RF	99.99%	23.71	1515.58
KNN	99.99%	106	1544.66
SVM	99.98%	2428.68	1792.39
DT	99.99%	2.40	1743.98
LR	99.96%	1.82	1754.58
Our method	99.92%	141	0.05

**Table 8 sensors-25-00216-t008:** The proposed model’s accuracy, the detection rate of a zero-day attack, false positive rate of the zero-day attack, and the average detection latency when some unseen attack classes from the three datasets are used as zero-day attack.

Zero-Day	Accuracy	DR	FPR	Detection Latency
NF-BoT-IoT				
Theft	99.62%	1.00	0.1496	0.0001
DDoS	99.70%	1.00	0.1356	0.0001
Recon	99.46%	1.00	0.0385	0.0001
NF-ToN-IoT				
Injection	99.84%	1.00	0.0037	0.0001
Malware	99.76%	1.00	0.0108	0.0001
DDoS	99.76%	1.00	0.0089	0.0001
NF-CSE-CIC-IDS 2018				
DDoS	99.92%	1.00	0.0037	0.0001
Malware	99.99%	1.00	0.0030	0.0001
Injection	99.99%	1.00	0.0032	0.0001

**Table 9 sensors-25-00216-t009:** The proposed model’s accuracy, detection rate of zero-day attack, false positive rate of the zero-day attack, and the average detection latency when unseen attack classes that contain synthetic zero-day attacks created using CTGAN from the three datasets are used as zero-day attacks.

Zero-Day	Accuracy	DR	FPR	Detection Latency
NF-BoT-IoT				
Theft	99.48%	1.00	0.1678	0.0001
DDoS	99.50%	1.00	0.1487	0.0006
Recon	99.16%	1.00	0.0671	0.0001
NF-ToN-IoT				
Injection	99.90%	1.00	0.0040	0.0001
Malware	99.82%	1.00	0.0119	0.0001
DDoS	99.88%	1.00	0.0097	0.0001
NF-CSE-CIC-IDS 2018				
DDoS	99.94%	1.00	0.0035	0.0001
Malware	99.96%	1.00	0.0030	0.0001
Injection	99.96%	1.00	0.0028	0.0001

**Table 10 sensors-25-00216-t010:** The model’s processing time, CPU usage (%), device memory usage (MB), and accuracy when the number of samples is increased.

# Samples	Time (s)	CPU (%)	Memory (MB)	Accuracy
NF-BoT-IoT				
50,000	24.25	12.40	45.84	98.10%
200,000	97.41	12.44	45.85	98.04%
350,000	171.12	12.45	45.85	98.13%
500,000	245.00	12.44	45.85	98.05
NF-ToN-IoT				
50,000	24.82	12.40	80.06	96.25%
200,000	98.41	12.47	80.06	96.22%
350,000	172.30	12.48	80.06	96.38%
500,000	245.99	12.49	80.06	96.42
NF-CSE-CIC-IDS 2018				
50,000	21.46	12.44	80.05	99.91%
200,000	85.99	12.45	80.06	99.92%
350,000	151.11	12.48	80.06	99.93%
500,000	216.30	12.38	80.06	99.96

**Table 11 sensors-25-00216-t011:** False positive rate evaluation under normal conditions.

# Normal Samples	FPR
NF-BoT-IoT	
13,858	0.00
NF-ToN-IoT	
179,646	0.00
NF-CSE-CIC-IDS 2018	
62,466	0.00

**Table 12 sensors-25-00216-t012:** The accuracy and F1 score of the proposed classifier when evaluated with NF BoT IoT dataset containing different kinds of Drift.

Type of Drift	Accuracy	F1
Gradual	90.95%	90.81%
Sudden	82.52%	82.38%
Recurring	87.66%	87.46%
Incremental	89.33%	88.97%

**Table 13 sensors-25-00216-t013:** The accuracy and F1 score of the proposed classifier when evaluated with NF ToN IoT dataset containing different kinds of Drift.

Type of Drift	Accuracy	F1
Gradual	89.55%	89.43%
Sudden	81.34%	81.20%
Recurring	86.31%	86.15%
Incremental	87.90%	87.69%

**Table 14 sensors-25-00216-t014:** The accuracy and F1 score of the proposed classifier when evaluated with NF CSE 2018 dataset containing different kinds of Drift.

Type of Drift	Accuracy	F1
Gradual	92.56%	92.56%
Sudden	83.77%	83.76%
Recurring	89.12%	89.11%
Incremental	90.81%	90.74%

**Table 15 sensors-25-00216-t015:** Comparing the mean CPU usage in percentage (%), the mean memory usage in kilobytes (KB), and the mean processing time in seconds (s).

Algorithm	CPU Usage (%)	Memory (KB)	Time (s)
NF-BoT-IoT			
SAMKNN	94.51	76.80	0.10
Adaptive SAMKNN	86.51	3.20	0.07
NF-ToN-IoT			
SAMKNN	90.27	548.20	0.10
Adaptive SAMKNN	83.61	51.20	0.07
NF-CSE-CIC-IDS 2018			
SAMKNN	87.25	619.20	0.10
Adaptive SAMKNN	81.12	70.40	0.07

**Table 16 sensors-25-00216-t016:** The Brier score of the proposed classifier when evaluated with each of the datasets.

Dataset	Brier Score
NF-BoT-IoT	0.0166
NF-ToN-IoT	0.0260
NF-CSE-2018	0.0028

**Table 17 sensors-25-00216-t017:** The t-statistic and *p*-value when the accuracy of the proposed classifier is compared with SAMKNN.

Dataset	t-Statistic	*p*-Value
NF-BoT-IoT	20.9729	0.0000
NF-ToN-IoT	24.1197	0.0000
NF-CSE-2018	8.5327	0.0010

**Table 18 sensors-25-00216-t018:** The accuracy, F1 score, processing time, and memory footprint when different components of the proposed classifier, such as adaptiveness, LTM, and SAM, are removed (ablation study).

Ablation Study	Accuracy	F1	Time (s)	Memory (KB)
NF-BoT-IoT				
Adaptive SAMKNN w/o LTM	98.09%	97.68%	110	73.91
SAMKNN with LTM	97.95%	97.39%	207	39,782.40
SAMKNN w/o LTM	98.09%	97.68%	93	33,024
KNN	97.74%	97.57%	911	5079.04
NF-ToN-IoT				
Adaptive SAMKNN w/o LTM	96.34%	96.23%	193	73.91
SAMKNN with LTM	95.76%	95.61%	344	59,002.88
SAMKNN w/o LTM	96.34%	96.23%	165	55,101.44
KNN	94.87%	94.80%	1528	5109.76
NF-CSE-CIC-IDS 2018				
Adaptive SAMKNN w/o LTM	99.92%	99.92%	194	73.91
SAMKNN with LTM	99.91%	99.91%	346	59,002.88
SAMKNN w/o LTM	99.92%	99.92%	165	55,101.44
KNN	98.96%	98.96%	1528	5038.08

## Data Availability

The code and the datasets used for our experimental validation are available in the GitHub repository, https://github.com/v-pragbe/Adaptive_Online_Attack_Detection.git.

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
