# Peer review of "A Scalable Approach to Internet of Things and Industrial Internet of Things Security: Evaluating Adaptive Self-Adjusting Memory K-Nearest Neighbor for Zero-Day Attack Detection"

_sensors, 2025, doi:10.3390/s25010216_

Round 1
Reviewer 1 Report
Comments and Suggestions for Authors
I grouped my revision in the following points:
1. Writing & Grammar
The manuscript is well-written and generally clear, but there are instances of verbose sentences and occasional grammatical errors. Some sections, such as the related work and methodology, are dense and could benefit from improved conciseness, especially in technical explanations (e.g., the Adaptive SAMKNN methodology). I also recommend the authors to use consistent terminology when referring to technical components (e.g., "Adaptive SAMKNN classifier" vs. "proposed classifier").
2. Title & Abstract
The title is descriptive and appropriate for the content. The abstract provides a strong overview but lacks sufficient details on the results and practical implications. This drives me to the recommendation for the authors to include key quantitative findings in the abstract (e.g., detection rates, memory savings). I would also like to recommend to add a sentence on the potential impact of the method for IoT and IIoT environments.
3. Figures, Tables, and Other Materials
Figures and tables are informative but lack standalone clarity due to insufficient captions. Some tables are dense and may overwhelm readers. I recommend the authors to consider splitting dense tables into smaller, more focused ones for better readability.
4. Problem or Question Addressed
The problem is clearly defined: addressing the challenges of zero-day attack detection in IoT and IIoT environments. However, I recommend the authors to emphasize why existing solutions are inadequate in the introduction, while highlighting the significance of addressing zero-day attacks in critical infrastructures.
5. Advance in the Field
The proposed Adaptive SAMKNN classifier represents a meaningful advancement in detecting zero-day attacks in resource-constrained IoT environments.
6. Originality
The study is original and builds upon prior work with significant enhancements to the SAMKNN classifier for IoT security. But still, I recommend the authors to explicitly discuss how this work differs from or improves upon the authors' previous conference paper.
7. Main Contributions
Contributions are well-defined and even listed in bullet points in the introduction.
8. Research Background
The manuscript provides an extensive background. The literature review is thorough and references relevant works, but there is a lack of critical comparison with competing methods. I recommend the authors to include a summary table comparing existing methods to the proposed approach.
9. Sources Citation
Citations are appropriate and sufficient, covering relevant studies.
10. Methodology
The methodology is detailed and logically presented, but some sections are overly complex. I suggest the authors to simplify explanations of the dynamic memory adjustment process for a broader audience.
11. Data Collection
Evaluation: The use of publicly available datasets is appropriate and well-justified. However, I could not find any description concerning the preprocessing steps for dataset preparation.
12. Sample Size
Evaluation: The sample sizes used in the experiments are large and suitable for scalability analysis.
13. Statistical Analysis or Qualitative Techniques?
The analysis is quantitative, relying on metrics like accuracy, F1 score, and false positive rate. The validity of the results could be strengthened if the authors decide to include confidence intervals or statistical tests.
14. Data Supporting the Results
Evaluation: Results are well-supported by experimental data, and the discussion aligns with findings.
15. Conclusion
The conclusion effectively summarizes the findings but lacks emphasis on practical implications and future work. I recommend the authors to add a section on real-world deployment scenarios and potential challenges.
16. Limitations
The manuscript briefly mentions the sensitivity of Adaptive SAMKNN to sudden drift. I recommend the authors to expand the limitations section to include constraints like dependency on specific datasets and resource requirements.
17. Future Research
Future research directions are briefly mentioned but lack depth. I recommend the authors to propose concrete steps for improving sudden drift detection and integrating with lightweight IoT devices.
Author Response
Thank you very much for taking the time to review this manuscript. Please find the detailed responses below and the corresponding revisions/corrections highlighted/in track changes in the re-submitted files.Reviewer#1, Concern # 1:
The manuscript is well-written and generally clear, but there are instances of verbose sentences and occasional grammatical errors. Some sections, such as the related work and methodology, are dense and could benefit from improved conciseness, especially in technical explanations (e.g., the Adaptive SAMKNN methodology). I also recommend the authors to use consistent terminology when referring to technical components (e.g., "Adaptive SAMKNN classifier" vs. "proposed classifier").
Author response: We thank the reviewer for this comment. We have revised the recommended sections by improving the technical explanations. We used “proposed classifier” as the consistent terminology.
Author action: The revision can be found throughout the manuscript. The technical explanation on page 9, lines 297-304, has been written simpler and more concisely.
Reviewer#1, Concern # 2:
The title is descriptive and appropriate for the content. The abstract provides a strong overview but lacks sufficient details on the results and practical implications. This drives me to the recommendation for the authors to include key quantitative findings in the abstract (e.g., detection rates, memory savings). I would also like to recommend to add a sentence on the potential impact of the method for IoT and IIoT environments.
Author response: We thank the reviewer for this comment. We have modified the abstract of our work to include key quantitative findings and added a sentence on the potential impact of our method for IoT/IIoT environments.
Author action: The modification can be found under “Abstract”, lines 8-10 and lines 16-19.
Reviewer#1, Concern # 3:
Figures and tables are informative but lack standalone clarity due to insufficient captions. Some tables are dense and may overwhelm readers. I recommend the authors to consider splitting dense tables into smaller, more focused ones for better readability.
Author response: Thank you for the comment. We have modified our captions to bring more clarity. We also split tables 3 and 8 in the original into smaller tables in the revised manuscript.
Author action: The revised manuscript has the captions of the following tables and figures modified to bring more clarity: Tables 4, 5, 6, 7, 8, 9, 10, 12, 13, 14 and 15. The captions of Figures 2, 3, and 4 were also modified.
Reviewer#1, Concern # 4:
The problem is clearly defined: addressing the challenges of zero-day attack detection in IoT and IIoT environments. However, I recommend the authors to emphasize why existing solutions are inadequate in the introduction, while highlighting the significance of addressing zero-day attacks in critical infrastructures.
Author response: Thank you for the comment. The original manuscript has a paragraph under the introduction elaborating why existing solutions are inadequate while highlighting the significance of addressing zero-day attacks in critical infrastructures.
Author action: This can be found on lines 45-56 under the “Introduction” section.
Reviewer#1, Concern # 5:
The study is original and builds upon prior work with significant enhancements to the SAMKNN classifier for IoT security. But still, I recommend the authors to explicitly discuss how this work differs from or improves upon the authors' previous conference paper.
Author response: We would like to thank the reviewer for this comment. This comment was worked on and the response is found under the “Introduction” section of the revised manuscript on lines 68-79.
Author action: We have explicitly explained how this work differs from the conference paper by stating quantitively the new work added. We also mentioned the components of the new work.
Reviewer#1, Concern # 6:
The manuscript provides an extensive background. The literature review is thorough and references relevant works, but there is a lack of critical comparison with competing methods. I recommend the authors to include a summary table comparing existing methods to the proposed approach.
Author response: Thank you for this comment. This comment has been worked on and can be found on pages 6 and 7 of the revised manuscript.
Author action: We have included a summary table comparing existing methods to our proposed method, as recommended.
Reviewer#1, Concern # 7:
The methodology is detailed and logically presented, but some sections are overly complex. I suggest the authors to simplify explanations of the dynamic memory adjustment process for a broader audience.
Author response: Thank you for the comment. The dynamic memory adjustment process has been simplified as recommended.
Author action: We have modified the dynamic memory adjustment process by presenting it in a simplified and concise language. This can be found on lines 297-304 of page 9 of the revised manuscript.
Reviewer#1, Concern # 8:
Evaluation: The use of publicly available datasets is appropriate and well-justified. However, I could not find any description concerning the preprocessing steps for dataset preparation.
Author response: We would like to thank the reviewer for the comment. We have acted on this comment by describing how the dataset was preprocessed.
Author action: We have described how the datasets were preprocessed. This description is found on pages 8 and 9, lines 218-283 of the revised manuscript.
Reviewer#1, Concern # 9:
The analysis is quantitative, relying on metrics like accuracy, F1 score, and false positive rate. The validity of the results could be strengthened if the authors decide to include confidence intervals or statistical tests.
Author response: Thank you for the comment. We added an additional experiment focusing on statistical analysis.
Author action: We addressed this comment by conducting statistical analysis using Bootstrap confidence interval, brier score, and paired t-test technique. This can be found on page 18, lines 630-649, and pages 27 and 28, lines 737-749.
Reviewer#1, Concern # 10:
The conclusion effectively summarizes the findings but lacks emphasis on practical implications and future work. I recommend the authors to add a section on real-world deployment scenarios and potential challenges.
Author response: Thank you for the comment. We highly appreciate it. We have added a subsection on real-world deployment scenarios and potential challenges.
Author action: We added a subsection on real-world deployment scenarios and challenges, which is found on pages 29 and 30, lines 869-892.
Reviewer#1, Concern # 11:
The manuscript briefly mentions the sensitivity of Adaptive SAMKNN to sudden drift. I recommend the authors to expand the limitations section to include constraints like dependency on specific datasets and resource requirements.
Author response: Thank you for this comment. We have included a subsection on limitations
Author action: We have revised the manuscript to include a subsection on limitations focusing on dependency on specific datasets and resource requirements. The limitations are found on page 29, lines 852-868.
Reviewer#1, Concern # 12:
Future research directions are briefly mentioned but lack depth. I recommend the authors to propose concrete steps for improving sudden drift detection and integrating with lightweight IoT devices.
Author response: Thank you for this comment. We have expanded the paragraph on future directions.
Author action: We have provided concrete steps for improving the detection of sudden drifts and integrating the proposed model with lightweight devices. This is found on page 30, lines 920-927.Reviewer 2 Report
Comments and Suggestions for Authors
This study investigates the effectiveness of Adaptive SAMKNN methods for detecting and responding to various attack types, including zero-day attacks, in the Internet of Things (IoT) and Industrial IoT (IIoT) environments. The work is interesting and appears significant in its context. However, some critical observations and suggestions for improvement are as follows:
Comment #1 (Contribution - describe what new contributions made):
As this work is an extended and revised version of the workshop paper [12], and the referenced work [12] has not yet been published, it is challenging to determine the extent of additional contributions made for this journal version. Please clearly specify and quantify the additional contributions (e.g., the percentage or amount of new work added) to ensure the journal paper provides significant value beyond the workshop version.
Comment #2 (Methodology - need more detailed description of modules):
The description of the modules in the methodology section (Section 3, Page 7) lacks sufficient clarity and detail. Each module should be defined comprehensively by addressing the following questions: What does the module do? Why is it included? How does it function? For example, in the Preprocessing subsection (Feature Extraction and Scaling), please elaborate on the features extracted, the rationale for selecting these features in the given context, and the methods used for extraction and scaling. This additional detail will improve the readability and overall clarity of the work.
Comment #3 (Zero-day attack detection - require detailed explanation of simulation):
On Page 11, the Zero-day Attack Detection scenario is described. The authors mention using a simulation approach for the experiments, which is critical but also highly sensitive to various factors. Two experiments were conducted: the first validated zero-day attacks using certain attack classes from the dataset, and the second employed a Conditional Tabular Generative Adversarial Network (CTGAN) to generate synthetic zero-day attacks. However, the description lacks sufficient clarity regarding how the simulations were conducted and how the CTGAN algorithm works. Additionally, no references are provided for the CTGAN method. Please include detailed explanations of the simulation processes and cite relevant references for the CTGAN technique.
Comment #4 (Evaluation metrics - Definition and relevance):
While several well-known metrics are used to evaluate the proposed methodology, their definitions and relevance to this work are not adequately described. Please provide clear definitions of the metrics used and explain why they are suitable for evaluating the performance of the proposed approach.
Author Response
Thank you very much for taking the time to review this manuscript. Please find the detailed responses below and the corresponding revisions/corrections highlighted/in track changes in the re-submitted files.
Reviewer#2, Concern # 1:
As this work is an extended and revised version of the workshop paper [12], and the referenced work [12] has not yet been published, it is challenging to determine the extent of additional contributions made for this journal version. Please clearly specify and quantify the additional contributions (e.g., the percentage or amount of new work added) to ensure the journal paper provides significant value beyond the workshop version.
Author response: We would like to thank the reviewer for this comment. This comment was worked on and the response is found under the “Introduction” section of the revised manuscript on lines 68-79.
Author action: We have explicitly explained how this work differs from the conference paper by stating quantitively the new work added. We also mentioned the components of the new work.
Reviewer#2, Concern # 2:
The description of the modules in the methodology section (Section 3, Page 7) lacks sufficient clarity and detail. Each module should be defined comprehensively by addressing the following questions: What does the module do? Why is it included? How does it function? For example, in the Preprocessing subsection (Feature Extraction and Scaling), please elaborate on the features extracted, the rationale for selecting these features in the given context, and the methods used for extraction and scaling. This additional detail will improve the readability and overall clarity of the work.
Author response: Thank you for this comment. We have modified modules in the methodology section to suit the recommendation.
Author action: We have provided a detailed description of the preprocessing steps (feature extraction and scaling), including the features extracted, the rationale for extracting such features, and the methods used. This is found on pages 8 and 9, lines 218-283.
Reviewer#2, Concern # 3:
On Page 11, the Zero-day Attack Detection scenario is described. The authors mention using a simulation approach for the experiments, which is critical but also highly sensitive to various factors. Two experiments were conducted: the first validated zero-day attacks using certain attack classes from the dataset, and the second employed a Conditional Tabular Generative Adversarial Network (CTGAN) to generate synthetic zero-day attacks. However, the description lacks sufficient clarity regarding how the simulations were conducted and how the CTGAN algorithm works. Additionally, no references are provided for the CTGAN method. Please include detailed explanations of the simulation processes and cite relevant references for the CTGAN technique.
Author response: Thank you for this comment. We have revised the manuscript based on this comment.
Author action: We have provided clarity on how the zero-day attack simulation was carried out in our experimental design. We have also provided a reference for CTGAN and provided a description of how CTGAN works. This is found on page 17, lines 576-589.
Reviewer#2, Concern # 4:
While several well-known metrics are used to evaluate the proposed methodology, their definitions and relevance to this work are not adequately described. Please provide clear definitions of the metrics used and explain why they are suitable for evaluating the performance of the proposed approach.
Author response: Thank you for this comment. We have provided definitions and relevance of the metrics used in our work.
Author action: We have provided clear definitions of the metrics used while explaining their relevance to our work and why they are suitable for evaluating the performance of the proposed technique. This is found on pages 14, 15, and 16, lines 482-557.
Round 2
Reviewer 2 Report
Comments and Suggestions for Authors
No comments with this second revised version!
Thanks.